# Brief communication : Evaluating Antarctic precipitation in ERA5 and CMIP6 against CloudSat observations

Marie-Laure Roussel [1], Florentin Lemonnier [1], Christophe Genthon [1], and Gerhard Krinner [2]

[1]Laboratoire de Météorologie Dynamique, Institut Pierre-Simon Laplace, Sorbonne Université / CNRS / École Normale Supérieure – PSL Research University / École Polytechnique – IPP, Paris, France
[2]Institut des Géosciences de l'Environnement, CNRS, Univ. Grenoble Alpes, 38000 Grenoble, France

**Correspondence:** (Marie-Laure Roussel (marie-laure.roussel@lmd.polytechnique.fr)

**Abstract.** CMIP5, CMIP6 and ERA5 Antarctic precipitations are evaluated against CloudSat data. At continental and regional scales, ERA5 and the median CMIP models are biased high, with insignificant improvement from CMIP5 to CMIP6. However, there are less positive outliers in CMIP6. AMIP configurations perform better than the coupled ones and, surprisingly, relative errors in areas of complex topography are higher (up to 50%) in the 5 higher resolution models. The seasonal cycle is well reproduced by the median of the CMIP models, but not by ERA5. Progress from CMIP5 to CMIP6 being limited, there is still room for improvement.

## 1 Introduction

Antarctica is the largest freshwater reservoir on Earth. Because of its sea-level equivalent of 57.9±0.9 m (Morlighem et al., 2019), even minor changes of the ice sheet mass balance can have important consequences for global sea level. Apart from a small contribution from ice deposition, precipitation is by far the dominant positive term in the ice sheet mass balance. At equilibrium it is compensated for by meltwater drainage and ice discharge (e.g., Favier et al., 2017). Precipitation is the main source of interannual mass balance variability of the ice sheet (e.g., Boening et al., 2012) and is projected to increase in a warmer future (e.g., Frieler et al., 2015). Therefore, an evaluation of the most recent CMIP6 (World Climate Research Programme (WCRP) Coupled Model Intercomparison Project phase 6) coordinated climate model simulations (Eyring et al., 2016) is timely.

Over the last decades, numerous technical developments have led to an increased number of meteorological measurements. In this study, precipitation over almost the entire Antarctic continent is analysed at a climatological time scale using a large-scale snowfall data-set that is independent from climate models, allowing objective evaluation. The reference for snowfall rate used here is the map produced by Palerme et al. (2014) based on the CloudSat satellite radar, which provided the first 4-year surface snowfall climatology for Antarctica. It has recently been followed by its complete three-dimensional version (Lemonnier et al., 2019b). We use these satellite observations to assess the Antarctic precipitation rates simulated by the CMIP6 models in various setups, at the continental and regional spatial scales, and at the annual and seasonal time scales. We further assess progress with respect to the preceding CMIP phase 5 (Taylor et al., 2012). ERA5 reanalyses are also used and evaluated in this comparison, because outputs are often used as a reference, particularly in less monitored areas, and because

of its foreseeable use as driver for regional climate models, the continental and climatological precipitation rates of which are strongly determined by the driving global model (e.g., Di Luca et al., 2012). Using new reanalyses and output of the most recent CMIP exercise, this work provides a brief update of the analysis by Palerme et al. (2017), which focused on CMIP5 and ERA-Interim.

## 2 Data and methods

### 2.1 Data

#### 2.1.1 Snowfall : CloudSat radar

The instrument on the CloudSat satellite platform is a RADAR operating at 94GHz and looking at nadir. The Cloud Profiling Radar (CPR) measures the back-scattered signal of hydrometeors. Based on micro-physical parameters (Wood et al., 2015) and the diffusion properties of the ice particles, the snowfall rate can be computed. Constrained by the satellite orbit, this
measurement can be performed up to 82°S. Many sources of error are related to this measurement: the various assumptions as well as the low frequency of passage of the satellite on the Antarctic induce uncertainties. (Lemonnier et al., 2019a) study allowed to improved confidence in the CPR snowfall retrieval over peripheral areas by a comparison with *in-situ* measurements (within maximum 25% error). In this work, we use data from the 2007-2010 Antarctic three-dimensional climatology (Lemonnier et al., 2019b) yielding the vertical distribution of the snowfall rate with a resolution of 1° latitude and 2° longitude
- optimizing the agreement with *in-situ* observations (Souverijns et al., 2018; Palerme et al., 2014). Recently the need to take into consideration the effect of soil echoes has been highlighted (Palerme et al., 2019), because it affects the measurement of CPR especially in the areas of complex topography, such as mountains and fjords. Some abnormal values are ignored in this dataset, but not highly impacting averages. Here we consider the radar information at the level of 1200 meters above ground level to assess the surface snowfall rate.

#### 2.1.2 CMIP5 and CMIP6 global climate models

The Coupled Model Intercomparison Project (CMIP, Taylor et al., 2012; Eyring et al., 2016) is coordinated by the World Climate Research Programme (WCRP). Its main objective is to improve modeling and future predictions combining the natural variability of the climate system and its response to modification of the radiative forcing in coordinated experiments (see https://es-doc.org/cmip6-experiments/). The available model outputs taken into account in this study are listed in table A1 of
50 the Appendix A. CMIP, which started in 1995, is currently in its $6^{th}$ phase.

Here we evaluate CMIP5 and CMIP6 model output from the *amip* and *historical* experiments. In the *amip* configuration, an atmospheric circulation model uses observed sea surface temperatures (SST) and sea ice (from 1979 to 2014) as prescribed boundary conditions. The so-called *historical* simulations are coupled ocean-atmosphere experiments. In both setups, observed time-varying atmospheric composition (anthropogenic, natural and volcanic influences), solar forcing, land use etc. based on
observations are prescribed. In addition, *highresSST-present*, defined in the framework of HighResMIP (Haarsma et al., 2016),

is a configuration available in the CMIP6 archive similar to *amip* with forced SST, but with a higher horizontal resolution. The experiment is designed to allow evaluating the sensitivity of climate model output to spatial resolution, and to help understanding the origins of model biases. The historical CMIP6 model outputs, driven by observed boundary conditions, end in 2014, while the observational period ended in 2005 in the earlier CMIP5 exercise. We therefore preferentially restrain the CMIP5 output to before 2005, complementing them by output from the RCP8.5 scenario run until 2014 where appropriate (see figure C1), because the realized $CO_2$ emissions between 2006 and 2014 closely follow those of that high emission scenario (Hayhoe et al., 2017). The start of our analysis period is 1979, corresponding to the beginning of the satellite period. We use all available CMIP5 and CMIP6 models, although it is well known (e.g., Masson and Knutti, 2011) that models managed by the same group or sharing a common development history yield very similar output, potentially biasing multi-model means. We preferentially use median model output, which is less sensitive to such effects, and quantify inter-model dispersion by the 25 and 75% percentiles, which are insensitive to outliers. Furthermore, although the *highresSST-present* multi-model ensemble of opportunity contains several versions of most models at low and high resolution, we do not restrain our choice to the high-resolution model versions; nevertheless, on average, the *highresSST-present* ensemble of opportunity used here has, on average, a substantially higher resolution than the *amip* and *historical* CMIP6 ensembles.

### 2.1.3 ERA5 reanalyses

ERA5 (Copernicus Climate Change Service (C3S), 2017) is the latest global reanalysis of the atmosphere made by the European Centre for Medium-Range Weather Forecasts (ECMWF) based on historical observation data since 1979 with the Integrated Forecasting System (IFS) model and its data assimilation system. Outputs from these reanalyses have high spatial horizontal and vertical resolutions (30 kilometers, 137 vertical levels). In this work, the monthly averages of the ERA5 reanalyses are used for the 40 years from 1979 to 2018.

### 2.2 Methods

For precipitation, we consider the entire Antarctic ice sheet, including ice shelves, where CloudSat satellite observations are available (i.e. north of 82°S). In order to evaluate the performances of the models in reproducing the various precipitation regimes of Antarctica, we examine both regional and seasonal averages. We consider the four standard meteorological seasons that are December-January-February (DJF), March-April-May (MAM), June-July-August (JJA) and September-October-November (SON). These are studied separately on the plateau (all areas above 2250 meters) and several peripheral and intermediate regions (defined by latitude and longitude, and an altitude below 2250 meters), as there are some seasonal signature differences mostly due to the sea-ice and the circumpolar current variations during the year with significant impact on precipitation patterns on the ice sheet margin (Palerme et al., 2017). Six regions have been selected based on latitude, longitude and altitude to distinguish main geographical patterns: Plateau, East Antarctic Coast, the Peninsula, the Filchner-Ronne and Ross Ice Shelves, and the remaining part of the West Antarctic Ice Sheet. These are shown in Figure 1 and described in Appendix B.

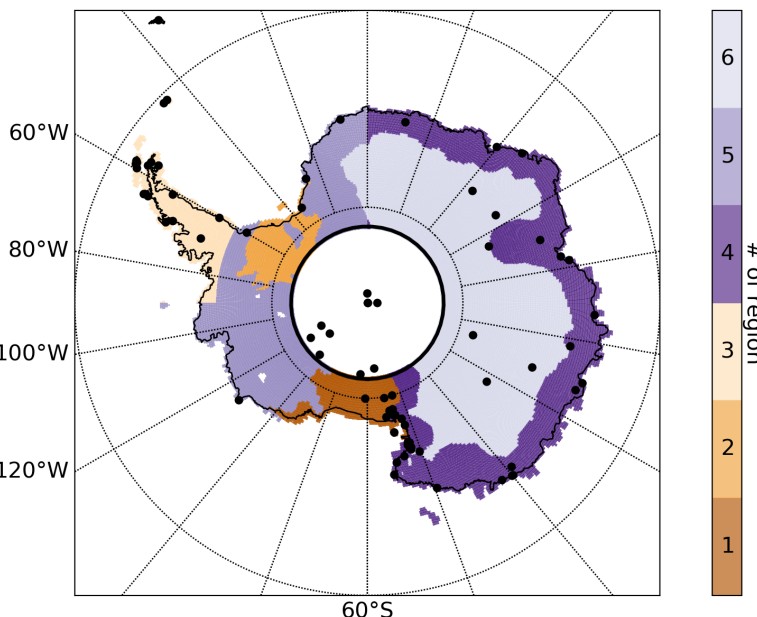

**Figure 1.** Map of the studied regions on the ERA5 grid. Numbers refer to the regions defined in table B1. Black dots indicate SCAR Reader temperature measurement stations (AWS and manned stations). The black line indicates the 82°S latitude circle.

To test the sensitivity of our conclusions concerning ERA and the CMIP outputs to the relatively short 4-year CloudSat period, we compare the CloudSat 4-year time series with multiple time periods of the same length extracted from the 40-year climatology of ERA5 and with the average of the 2007-2010 CloudSat period. We made 20 draws of 4 random years to process the samples for the evaluation against the 2007-2010 CloudSat period. This number of 20 samples has been chosen because there is no significant difference in the results with more samples. As we will show below (see section 2.3.1), our conclusions are not very sensitive to these choices.

Furthermore, as historical CMIP5 outputs are only available for years up to 2005, a direct comparison from 2007 to 2010 is not possible between CMIP5 and CloudSat. Annual mean snowfall (averaged over the whole Antarctic continent north of 82°S) starting in 1979 is available until 2005 for CMIP5, until 2014 for CMIP6, and until 2018 for ERA5. Over this period, there is a slight positive mean precipitation trend in the CMIP ensembles (strongest, about 2% per decade, in the CMIP5 and CMIP6 *historical* simulations), but the variations induced by this trend over the model periods are substantially weaker than the absolute differences between the model means and the CloudSat observational average. Therefore, and because our results are not particularly sensitive to the choice of model years, CMIP output is averaged over the entire respective simulation period for comparison with CloudSat.

A more detailed statistical analysis using the Welch t-test, presented in Appendix D, demonstrates that the greatest confidence is attached to the ice-shelves, the Peninsula and the East coast (Ross, Filchner, Peninsula and Low East regions) when

comparing the snowfall means from CloudSat to ERA5 and the CMIP (both 5 & 6) datasets. However, some uncertainties remain in areas of complex topography, due to sublimation of snow below the first level of CloudSat, which is likely to influence the total snowfall amount taken into account here. In the interior of the Antarctic continent, the comparison has to be treated with caution as the snowfall means from CloudSat, ERA5 and the CMIP datasets are significantly different. This may be mainly due to the CPR of CloudSat that underestimates snowfall means, as a major part of it comes from microphysical processes occuring below the first CloudSat level in this region. Therefore, the comparison is focused on the differences between CMIP5 and CMIP6, while the CloudSat results are kept for information purposes only as the single source of observation over these areas. In addition, the test points out that there is no major reduction of the reliability of the comparison between CloudSat and the CMIP experiments when the whole temporal coverage is considered (instead of a 4-years time-series). Conversely, there is a more significant influence of the selected years of the ERA5 dataset, which is more sensitive to the inter-annual variability.

## 2.3 Results

### 2.3.1 Continent-wide climatological snowfall rates

Figure 2 displays the annual precipitation for the entire continent ("All") and the defined regions for CloudSat, ERA5 (both the 2007-2010 period and the average of 20 draws of four random years with associated standard deviation) and the various CMIP5 and CMIP6 ensembles. For all CMIP experiments, the ensemble median of the continental mean precipitation is above the 2007-2010 CloudSat average of 186 mm water equivalent per year. Following Palerme et al. (2017) who compared CMIP5 models to CloudSat snowfall measurements, we have identified CMIP5 and CMIP6 models that have continent-wide mean snowfall rates within 20% of the CloudSat average value of 186 mm water equivalent per year, that is, between 150 and 223 mm per year. Not a single CMIP5 and CMIP6 model falls below this lower bound. Conversely, a substantial fraction of CMIP models, both in CMIP5 and CMIP6, exceeds the upper bound of 223 mm per year. As a result, only 58% of the CMIP6 *amip* models fall within the ±20% range around the CloudSat value, and this number decreases to 38% for CMIP6 *highresSST-present*, the other ensembles lying between these extreme values. The atmosphere-only *amip* runs less frequently exceed the ±20% bound (56% and 58% within the 20% range for CMIP5 and CMIP6, respectively) than the coupled *historical* runs (43% and 48% within the 20% range for CMIP5 and CMIP6, respectively). We must note that the median model precipitation rate shows no improvement from CMIP5 to CMIP6; if anything, compared to CMIP5, there is even a degradation in the CMIP6 median *historical* simulation with respect to CloudSat.

There is therefore a systematic high bias, exacerbated a higher spatial resolution, and no substantial improvement is obvious on the continental scale from CMIP5 to CMIP6; prescribed observed oceanic boundary conditions (SST and sea ice) in the *amip* runs lead, unsurprisingly, to more realistic simulated precipitation rates than in the corresponding coupled runs.

From CMIP5 to CMIP6, one can note, on the positive side, that the number of models with extreme positive precipitation biases is reduced. In the CMIP5 *historical* ensemble, for example, 4 models exceed (in one case very substantially) the maximum of the CMIP6 ensemble at 353 mm, which is almost twice the observed 2007-2010 rate.

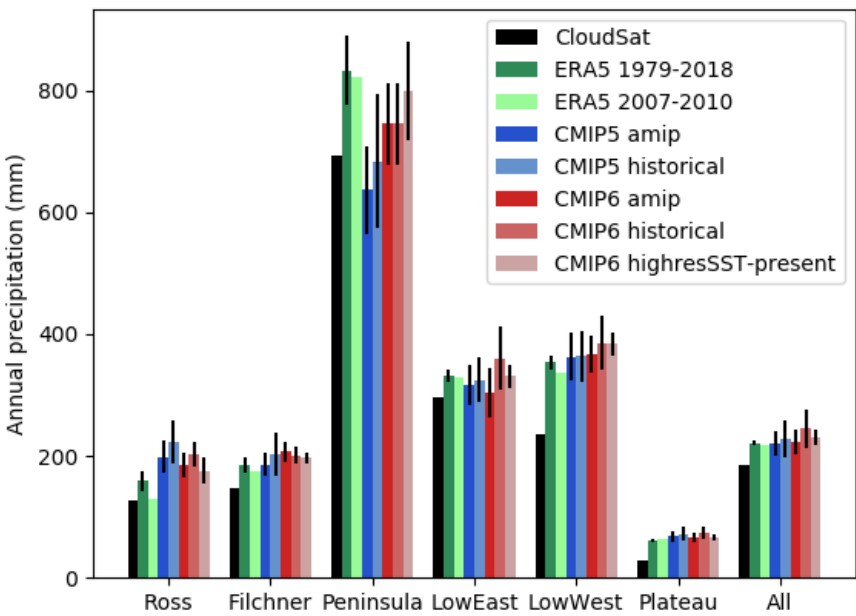

**Figure 2.** Average snowfall rate per region and for the entire continent north of 82°S for CloudSat (black, 2007-2010), ERA5 (computed on 20 draws of 4 random years between 1979 and 2018, with standard deviation, and the 2007-2010 average), and the various CMIP ensembles. For CMIP, the ensemble medians and the 25[th] and 75[th] percentiles are indicated.

Interestingly, ERA5 similarly exhibits a positive mean precipitation bias of about 30 mm per year, and is therefore not better, at least compared to the CloudSat climatology, than the CMIP5 and CMIP6 median models.

### 2.3.2 Regional averages

Figure 2 shows that ERA5 and the CMIP6-highresSST models, which have higher horizontal resolutions that should enable a better spatial representation of the small scale processes, particularly those induced by topography, do not exhibit reduced errors in the Peninsula region and in West Antarctica (regions named LowWest, Filchner and Ross). Relative errors with respect to the CloudSat measurement can exceed 50% in these regions, compared to the lower regions of East Antarctica (LowEast) where it is as low as a few percents.

Conspicuously, all CMIP ensembles and ERA5 exhibit positive biases with respect to CloudSat in all regions. The strongest relative biases are located in the Plateau region, that is, above 2250 m, where theCloudSat mean is about 29 mm of water equivalent per year, while the ERA mean for the same period is 65 mm per year, and the CMIP ensembles have even stronger

biases. In most regions, the *amip* simulations exhibit lower biases than the coupled *historical* simulations in the CMIP5 and CMIP6 ensembles, as already seen for the continental mean values.

There is no clear overall improvement in the performance of the CMIP6 ensemble over the CMIP5 ensemble. There is degradation in some regions (for example the Peninsula) and improvement in others, such as the Plateau region, where the improvement in the *amip* configuration is modest (see also Figure 3), but important because of the large spatial extent of the East Antarctic Plateau, and on the Ross Ice Shelf. In these plateau and ice shelf areas, the *highresSST-present* runs consistently perform better than the other CMIP6 runs. This is contrary to expectations that higher spatial resolution, by leading to a better representation of topographical effects, would in principle allow better representing precipitation rates in regions with steep topography, that is, mostly coastal areas.

### 2.3.3  Seasonal averages

Figure 3 displays the observed and simulated seasonal variations of precipitation separately for the high (>2250 m) and low (<2250 m) regions of the continent. The CMIP ensembles capture the weak annual cycle in the plateau regions, characterized by a maximum in DJF and a minimum in SON, but, as reported above, they overestimate the average precipitation rate substantially. ERA5 does not capture this seasonality and simulated maximum precipitation rates in MAM and JJA. In the lower reaches of the continent, the CMIP ensembles and ERA5 do capture the observed seasonality, with maximum precipitation rates typically in MAM. This is very probably linked to the availability of oceanic moisture, driven by sea ice around the continent and the delayed annual temperature cycle in the Southern Ocean, and to the seasonality of meriodional atmospheric circulation (Genthon and Krinner, 1998).

### 3  Discussion and conclusion

The CloudSat precipitation climatology provides the possibility to evaluate climate models and reanalyses against model-independent satellite-derived data. By comparing ERA5 reanalysis output from multiple random 4-year periods against output for the 4-year observational period (2007-2010) and the satellite-derived data, we have shown that on regional scales, a 4-year period is long enough to draw robust conclusions about misfits between the models and the satellite data set.

The main results of this short study are that: 1) All CMIP model ensemble medians and ERA5 overestimate the continental mean precipitation rates; 2) The positive biases are particularly strong in the plateau regions; 3) There is no measurable improvement, in terms of continental and regional mean precipitation rates and their seasonality, from CMIP5 to CMIP6; 4) The seasonal cycle of precipitation, both on the plateau and lower (coastal) regions, is reasonably well captured by the median CMIP models; 5) Median precipitation rates tend to be better reproduced in the atmosphere-only *amip* configurations than in the coupled *historical* setups; 6) Positive precipitation biases in particular in the Peninsula region are exacerbated at higher resolution in the *highresSST-present* ensemble; 7) The CMIP6 ensemble suffers less than CMIP5 from outliers with very strong positive precipitation biases.

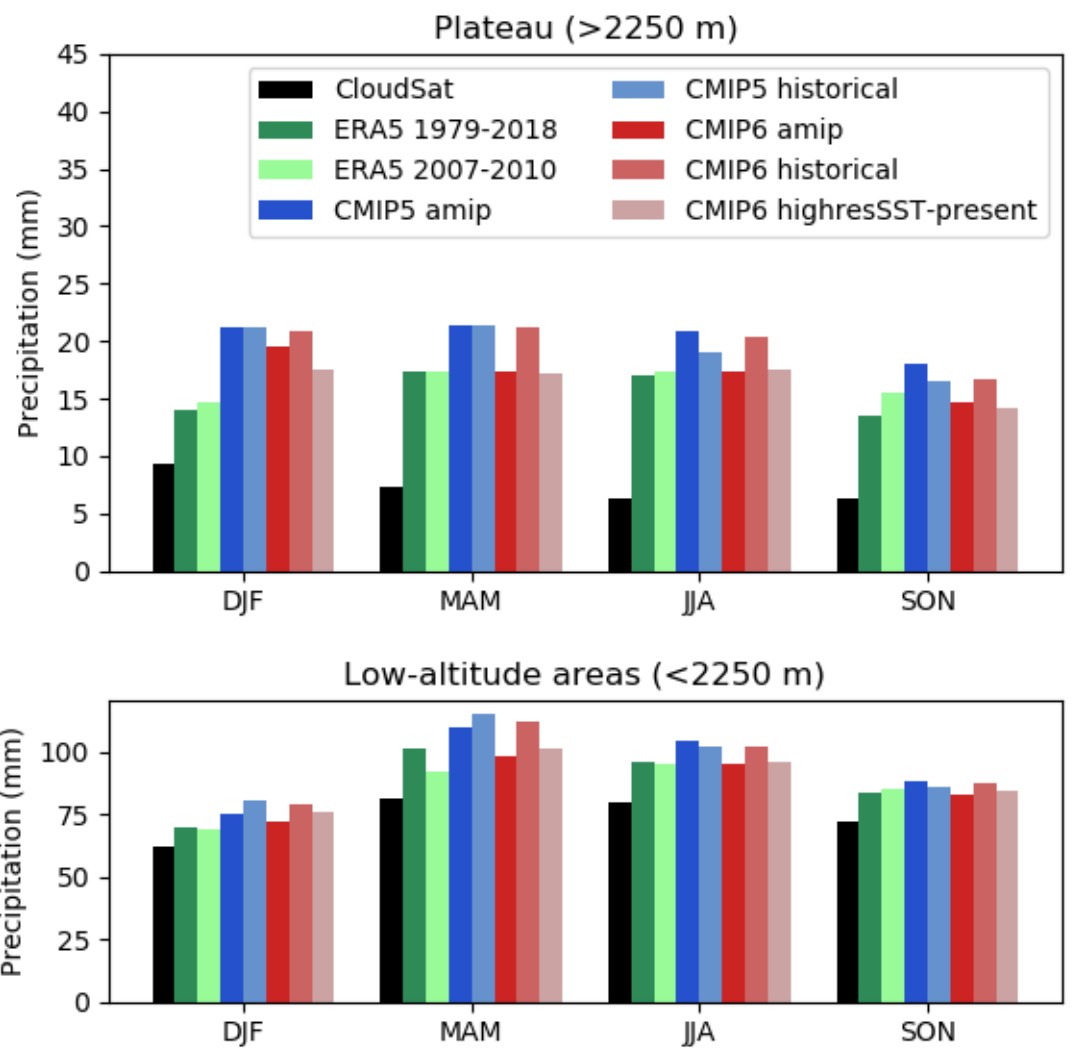

**Figure 3.** Seasonal averages of the observed mean 2007-2010 CloudSat, mean ERA 5 (random and 2007-2010 averages), and ensemble median long-term average CMIP5 and CMIP6 snowfall rates per season for the plateau areas (top) and the low-lying reaches of the continent (bottom).

We note that although there is no progress in the representation of large-scale mean precipitation and of its seasonality from CMIP5 to CMIP6, there is a concomitant slight progress in the representation of surface air temperature. Regional-scale

multi-model median root-mean square errors are reduced by typically 5 to 10% between these successive CMIP generations (see Figure E1 in the annex). This indicates that in spite of a clear physical link between temperature and precipitation changes on long time scales (e.g., Frieler et al., 2015), precipitation errors in current-generation AGCMs are not dominated by the first-order physical link between temperature and water vapour saturation pressure, but by errors in the representation of other processes such as atmospheric circulation and cloud microphysics.

*Author contributions.* This research was designed by all authors. MLR and GK carried out the data analysis. MLR wrote the initial draft and all coauthors contributed to the writing.

*Competing interests.* The authors declare no competing interests.

*Acknowledgements.* This paper is a contribution to the CALVA and APRES3 projects supported by the French polar institute IPEV. Support by the French space agency CNES for program EECLAT is also acknowledged. MLR benefits a PhD grant by Ecole Polytechnique. We acknowledge the World Climate Research Programme, which, through its Working Group on Coupled Modelling, coordinated and promoted CMIP5 and CMIP6. We thank the climate modeling groups for producing and making available their model output, the Earth System Grid Federation (ESGF) for archiving the data and providing access, and the multiple funding agencies who support CMIP6 and ESGF.

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

## Appendix A: CMIP5 & CMIP6 version models

| CMIP5 | |
|---|---|
| *amip* | ACCESS1-0, ACCESS1-3, bcc-csm1-1-m, bcc-csm1-1, BNU-ESM CCSM4, CESM1-CAM5, CMCC-CM, CNRM-CM5, CSIRO-Mk3-6-0, EC-EARTH, FGOALS-g2, FGOALS-s2, GFDL-CM3, GISS-E2-R inmcm4, IPSL-CM5A-LR, IPSL-CM5A-MR, IPSL-CM5B-LR, MIROC-ESM MIROC5, MPI-ESM-LR, MPI-ESM-MR, MRI-CGCM3, NorESM1-M |
| *historical* | ACCESS1-0, ACCESS1-3, bcc-csm1-1-m, bcc-csm1-1, BNU-ESM, CanCM4, CanESM2, CCSM4, CESM1-BGC, CESM1-CAM5, CESM1-FASTCHEM, CESM1-WACCM, CMCC-CESM, CMCC-CMS, CNRM-CM5-2, CNRM-CM5, CSIRO-Mk3-6-0, CSIRO-Mk3L-1-2, EC-EARTH, FGOALS-g2, FGOALS-s2, FIO-ESM, GFDL-CM2p1, GFDL-CM3, GFDL-ESM2G, GFDL-ESM2M, GISS-E2-H-CC, GISS-E2-H, GISS-E2-R-CC, GISS-E2-R, HadCM3, HadGEM2-AO, HadGEM2-CC, HadGEM2-ES, inmcm4, IPSL-CM5A-LR, IPSL-CM5A-MR, IPSL-CM5B-LR, MIROC-ESM-CHEM MIROC-ESM, MIROC4h, MIROC5, MPI-ESM-LR, MPI-ESM-MR, MPI-ESM-P, MRI-CGCM3, MRI-ESM1, NorESM1-M, NorESM1-ME |
| CMIP6 | |
| *amip* | BCC-CSM2-MR, BCC-ESM1, CAMS-CSM1-0, CanESM5, CESM2-WACCM CESM2, CNRM-CM6-1, CNRM-ESM2-1, E3SM-1-0, EC-Earth3-Veg, EC-Earth3, FGOALS-f3-L, GFDL-CM4, GFDL-ESM4, GISS-E2-1-G, HadGEM3-GC31-LL, INM-CM5-0, IPSL-CM6A-LR, MIROC6, MRI-ESM2-0, NESM3, NorCPM1, NorESM2-LM, SAM0-UNICON, UKESM1-0-LL |
| *historical* | BCC-CSM2-MR, BCC-ESM1, CAMS-CSM1-0, CanESM5, CESM2-WACCM, CESM2, CNRM-CM6-1-HR, CNRM-CM6-1, CNRM-ESM2-1, E3SM-1-0, EC-Earth3-Veg, EC-Earth3, FGOALS-g3, GFDL-CM4, GFDL-ESM4, IPSL-CM6A-LR, GISS-E2-1-G, GISS-E2-1-H, HadGEM3-GC31-LL, INM-CM4-8, MIROC-ES2L, MIROC6, MRI-ESM2-0, NESM3, NorCPM1, NorESM2-LM, SAM0-UNICON, UKESM1-0-LL |
| *highresSST* | CMCC-CM2-HR4, CMCC-CM2-VHR4, CNRM-CM6-1-HR, CNRM-CM6-1, ECMWF-IFS-HR, ECMWF-IFS-LR, FGOALS-f3-H, FGOALS-f3-L, GFDL-CM4C192, HadGEM3-GC31-HM, HadGEM3-GC31-LM, HadGEM3-GC31-MM, INM-CM5-H, IPSL-CM6A-ATM-HR, IPSL-CM6A-LR, MPI-ESM1-2-HR, MPI-ESM1-2-XR, MRI-AGCM3-2-H, MRI-AGCM3-2-S, NICAM16-7S, NICAM16-8S |

**Table A1.** CMIP5 and CMIP6 models considered in this study

## Appendix B:  Geographical delimitations for the regional analysis

| Regions | 1 : Ross | 2 : Filchner | 3 : Peninsula | 4 : LowEast | 5 : LowWest | 6 : Plateau |
|---|---|---|---|---|---|---|
| Latitude | -99°,-75° | -99°,-76° | -74°,-59° | -99°,-59° | -99°,-59° | -99°,-59° |
| Longitude | 150°,240° | 270°,340° | 270°,320° | 0°,180° | 180°,360° | 0°,360° |
| Altitude | < 300m | < 300m | - | < 2250m | < 2250m | > 2250m |
| Number of stations | 15 | 1 | 19 | 39 | 30 | 9 |

**Table B1.** Selection criteria applied to define the studied regions.

**Appendix C:  CMIP5, CMIP6, ERA5 and CloudSat time series of mean annual surface precipitation rates**

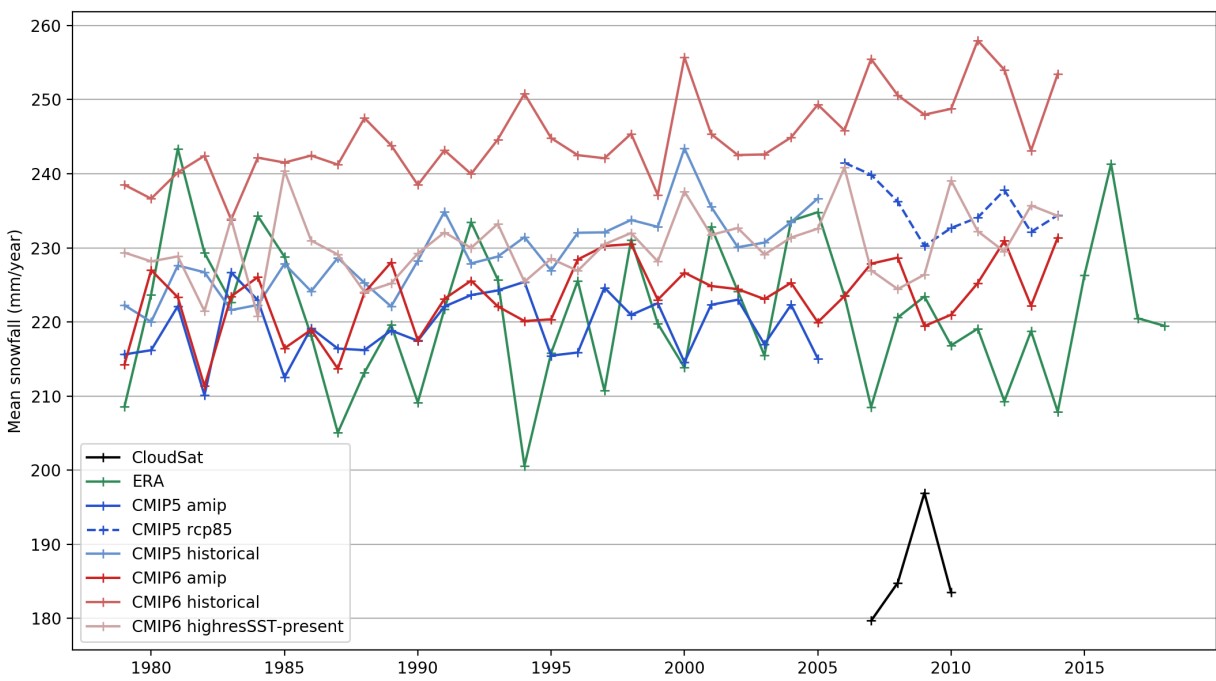

**Figure C1.** 1979 to 2018 time-series of mean annual snowfall rate over the entire Antarctic continent. The median values for all the models of each CMIP (5 & 6) and the mean values for CloudSat and ERA5 are plotted.

**Appendix D: Statistical significance of the 4-year CloudSat snowfall climatology**

Snowfall means of the CloudSat dataset have been compared to ERA5 ones and to the median of each of the CMIP (5 & 6) experiments with a Welch's t-test. This statistical tool is used to assess the difference between the mean of two independent groups that have normal distributions with unequal variances. Those assumptions have been verified by all the datasets at the continent scale and for the whole year. The hypothesis tested (null-hypothesis) is that the means of the two groups are equal. We performed the two-tailed test and analysed the p-value (rounded down to one hundredth) - in comparison with the

0.05 threshold - for each region (defined in the Table B1) and each season. Three methods have been carried out to chose the time-serie of the datasets used in this test, illustrated by the Figure D1 :

–    whole temporal coverage : the entire time-serie available (respectively 1979-2018 for ERA5, 1979-2005 for CMIP5 and 1979-2015 for CMIP6)

–    correct years : the exact years of the CloudSat climatology (only for ERA5 and the three CMIP6 experiments)

–    random years : draws of 4 non-consecutive years (tested from 1 to 10000 draws and limited to 20 draws in the main work)

p-values are generally higher for the first 4 regions (Ross, Filchner, Peninsula and Low East) for any season and for both ERA5 and CMIP (5 & 6). The choice of the time-serie has no major impact on the result of the test for each of the CMIP experiments. On the contrary ERA5 and CloudSat are in a much better agreement when considering a 4-years time-serie. The

Figure D2 presents the detailed results considering the whole time coverage for each of the CMIP experiment and for the three time-series considered for ERA5. The red color indicates when the null-hypothesis has to be rejected and the blue color when it can not be rejected. One can note that the snowfall mean is significantly different at the continent scale for any season, as well as on the plateau and the west coast. Higher p-values are mainly on ice-shelfes, peninsula and the east coast.

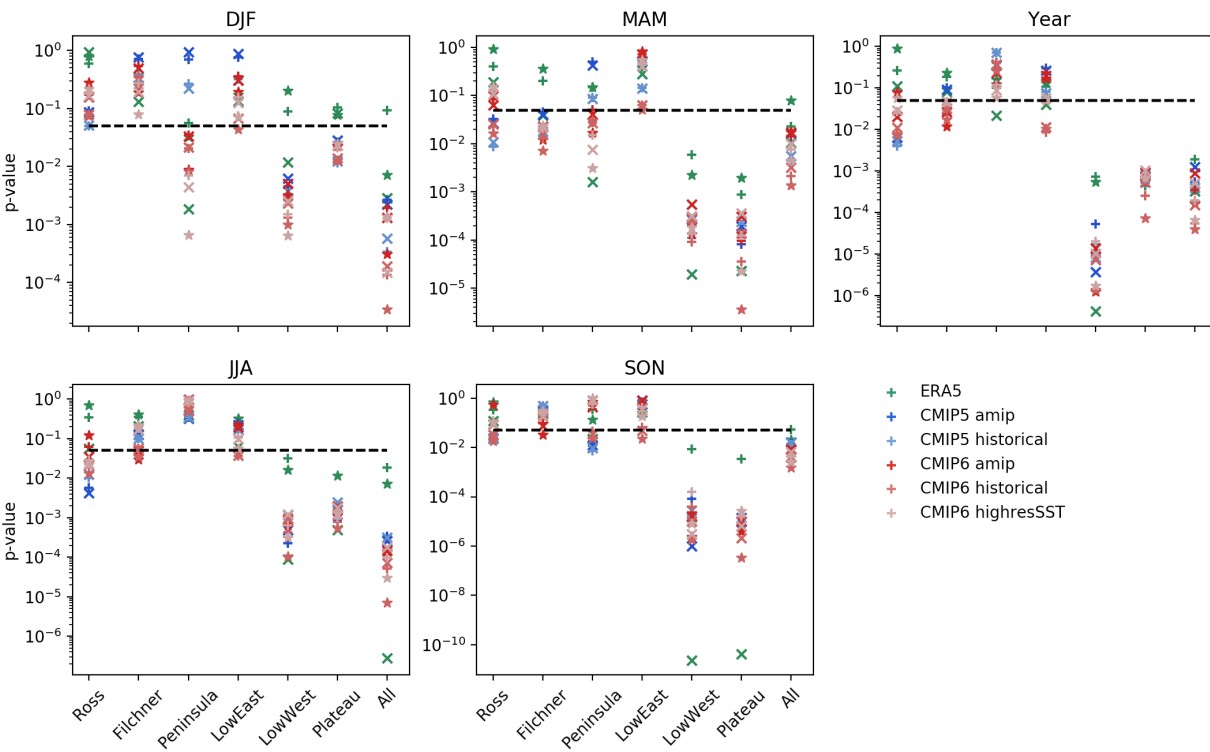

**Figure D1.** p-values of the Welch t-test comparing the snowfall means from CloudSat to ERA5 and each of the CMIP (5 & 6) experiments for various seasons and the whole year on each of the regions considered. The black dashed line shows the 0.05 threshold to decide whether the hypothese is rejected or not. 'x' crosses indicate results considering the whole temporal coverage, '+' crosses the random 4-years and stars the correct 4-years of the CloudSat climatology.

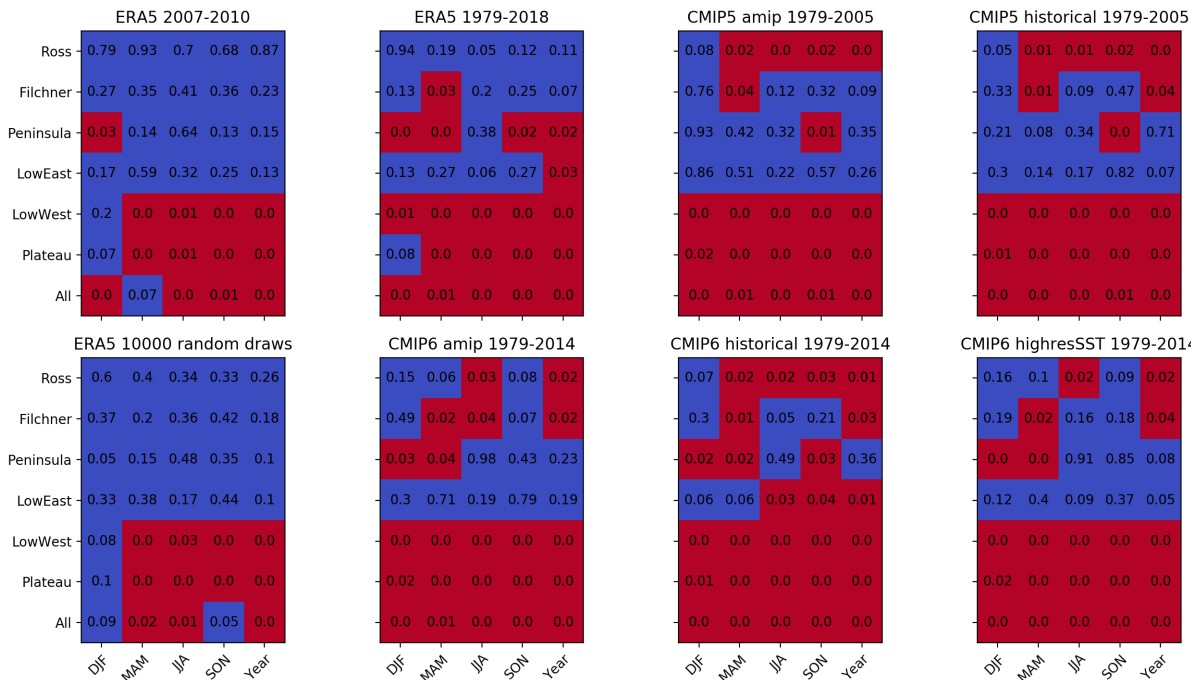

**Figure D2.** p-values of the Welch t-test comparing the snowfall means from CloudSat to ERA5 and each of the CMIP (5 & 6) experiments for various seasons and the whole year on each of the regions considered, considering the whole temporal coverage. Blue color indicates that the p-value is greater then the 0.05 threshold (null-hypothesis can not be rejected), red color indicates that the p-value is lower than the threshold (null-hypothesis rejected).

## Appendix E: CMIP5 & CMIP6 surface air temperature comparison to SCAR Reader stations

Changes in the quality of the representation of observed precipitation rates are briefly assessed in the light of temperature biases with respect to SCAR READER (REference Antarctic Dataset for Environemental Research) AWS (Automatic Weather Station) and manned station data (Turner et al., 2004). For each station and model, we identified the nearest grid point and used a spatial regression (based on the neighboring grid points) of surface temperature against surface altitude in order to correct for altitude differences between the model and the observations. SCAR Reader data were used only when at least 10 years of

observations were available, and the model output was averaged over the number of years of available observations, centered around the mean year of these observations between 1979 and 2005 (in order to evaluate progress from CMIP5 to CMIP6).

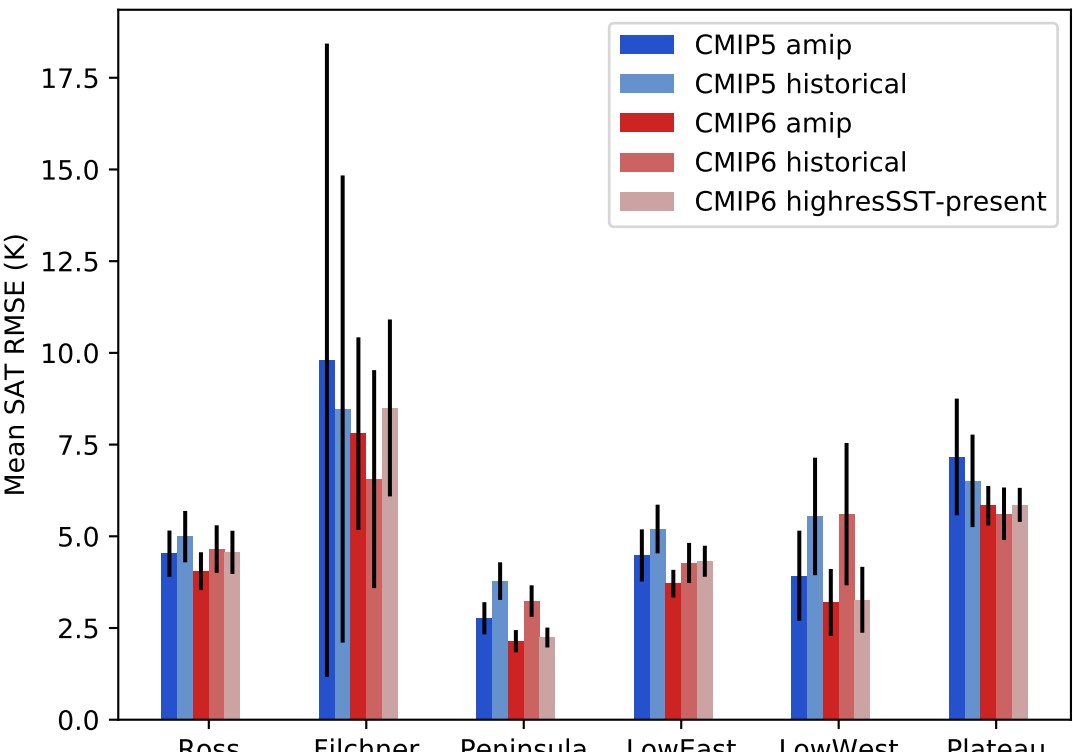

**Figure E1.** Multi-model mean of the multi-station mean root-mean square error (RMSE, in K) of simulated monthly surface air temperatures against SCAR Reader stations (AWS and manned), for the different regions. The regional mean inter-model standard deviation is shown as black error bars, indicating, for some regions, reduced inter-model spread in CMIP6 compared to CMIP5 and modest overall improvement.