# Peer review of "Brief communication : Evaluating Antarctic precipitation in ERA5 and CMIP6 against CloudSat observations"

_The Cryosphere, 2019_

## Referee Comment (RC1) · John King (Referee) · 6 Mar 2020

Review of "Evaluating Antarctic precipitation in ERA5 and CMIP6 against CloudSat observations", by Roussel et al (tc-2019-327)

General comments

In this useful short paper, the authors use a new climatology of Antarctic precipitation derived from CloudSat measurements to assess the representation of precipitation in the ERA5 reanalysis and the CMIP5 and CMIP6 climate model ensembles. They find that, relative to CloudSat, all models overestimate Antarctic precipitation and that there

has been no significant improvement between CMIP5 and CMIP6. A somewhat surprising finding is that the higher-resolution subset of the CMIP6 ensemble appears to perform worse than the lower-resolution simulations over regions of complex orography such as the Antarctic Peninsula. This clearly warrants further investigation, but is outside the scope of the current paper.

The CloudSat dataset covers a relatively short time period (4 years), which raises questions about its representativity. However, the authors use a Monte Carlo approach to demonstrate that this is not a problem. With a little more work it might be possible to use this same approach to make a quantitative assessment of the statistical significance of the differences between models and observations.

I recommend publication of this paper following minor revisions.

Specific comments and technical corrections

Please see the attached annotated manuscript file.

Please also note the supplement to this comment:
https://www.the-cryosphere-discuss.net/tc-2019-327/tc-2019-327-RC1-supplement.pdf

─────────────────────────────

**Supplement:**

[revised manuscript text omitted]

---

## Referee Comment (RC2) · Anonymous Referee #2 · 8 Apr 2020

This 'brief communication' paper presents a comparison of Antarctic precipitation rates as derived by CloudSat, and as simulated by ERA5 and CMIP5 and CMIP6 climate models. The results indicate that there are substantial biases in ERA5 and CMIP models, and show no clear improvement from CMIP5 to CMIP6. While this is relevant work and a potentially interesting work, I am not sure why the authors chose to turn it into a 'brief communication' paper, given that I think many results are somewhat unsubstantiated and/or incomplete, and important details are missing. Below, I'd like to highlight two crucial items that I consider prohibitive for publication of this work, and would need to be worked on in case the authors would like to revise the paper. I will not focus on minor/textual items at this point.

**CloudSat quality:** While the authors present CloudSat as a benchmark data set, its performance over the Antarctic interior is highly doubtful. Across the ice sheet interior, a large fraction of annual the snowfall is generated within the shallow (stable) boundary layer by vapor (re)deposition and/or diamond dust. Since the depth of this inversion layer is typically (much) shallower than 1200 m, CloudSat fails to detect much of the precipitation in the interior. Recent studies have focused comparison with in-situ radars and other products in the coastal or escarpment areas (e.g. Lemonnier et al., 2019; Souverijns et al., 2018). The fair agreement of CloudSat with ERA-Interim as presented in Palerme et al., 2014 (TC) is associated with the well-documented dry bias in ERA-Interim in the interior (e.g. Medley and Thomas, 2019 (Nature Clim. Change)), which is confirmed here to some extent with ERA5. Based on this, I think the authors cannot claim that CloudSat is reliable in areas >2250 m (or even lower), and this analysis should be removed. The authors could instead focus on coastal areas only, where CloudSat is likely performing better, although uncertainties are likely still substantial where topography is complex and where sublimation of suspended snow particles can play a role between 0 and 1200 m (Grazioli et al., 2017 (PNAS)).

**Temperature and precipitation:** the authors aim to relate the bias in precipitation to temperature biases, suggesting that although the temperature is improved in CMIP6 relative to CMIP5, precipitation is not. I think this analysis falls short in multiple aspects:
   (a) As the authors surely can confirm, there are many levels of complexity involved with precipitation formation, and only focusing on near-surface temperature definitely understates these complexities. While there might be some sort of relation between the two, the authors should focus on many other aspects of precipitation process (e.g. cloud microphysics, atmospheric (thermo)dynamic structure, humidity, etc. etc.). I simply don't think it is acceptable to argue that model performance in near-surface temperature and precipitation are related.
   (b) The statistical analysis of near-surface temperature needs to be expanded. Only the RMSE is currently shown, but that fails to represent the mean bias (i.e. does the mean bias improve between CMIP5 and CMIP6?). Moreover, there is no analysis of statistical significance whatsoever, which is clearly necessary if the authors want to claim that temperature representation has improved in CMIP6.

One smaller issue is that Figure 1 should mask out regions south of 82 South. What are the white areas.

---

## Author Comment (AC1) · 7 May 2020

**Replies to Dr. John King**

May 7, 2020

**Performance of the higher-resolution subset of the CMIP6 ensemble**

- Reviewer comment: *A somewhat surprising finding is that the higher-resolution subset of the CMIP6 ensemble appears to perform worse than the lower-resolution simulations over regions of complex orography such as the Antarctic Peninsula. This clearly warrants further investigation, but is outside the scope of the current paper.*

- Reply: We fully agree that this point is of particular interest. It will requires a special analysis in a future dedicated work since this is not the focus of this article.

**Representativity of the CloudSat measurements**

- Reviewer comment: *The CloudSat dataset covers a relatively short time period (4 years), which raises questions about its representativity. However, the authors use a Monte Carlo approach to demonstrate that this is not a problem. With a little more work it might be possible to use this same approach to make a quantitative assessment of the statistical significance of the differences between models and observations.*

- Reply: A statistical analysis has been added in Appendix D to distinguish the regions where the comparison with the CloudSat dataset can be trusted with a good confidence level. It points out that the best reliability is on coastal regions, ice-shelves and peninsula, whereas there is a significant difference between the CloudSat snowfall means and re-analyses and model results on the interior of the continent. In addition, we carried out the same test with various time-series (whole temporal coverage and selected or random 4-years). The conclusion is that this choice has no significant influence on the result of the statistical test

for the CMIP (5 & 6) experiments whereas it plays a major role on ERA5 performances.

**Other remarks**

The number of stations per region has been added to the Table B1 in Appendix B. Missing acronyms have been detailed. The color palette has been modified for all figures to match the ones used by the IPCC for the CMIP (5 & 6) experiments.

---

## Author Comment (AC2) · 7 May 2020

**Replies to Reviewer 2**

May 7, 2020

**Scope and format of this Brief Communication**

- Reviewer comment: *While this is relevant work and a potentially interesting work, I am not sure why the authors chose to turn it into a 'brief communication' paper, given that I think many results are somewhat unsubstantiated and/or incomplete, and important details are missing.*

- Reply: This Brief Communication should be considered as an undate to the article "Evaluation of current and projected Antarctic precipitation in CMIP5 models" by Palerme et al. (*Climate Dynamics*, 2017, doi:10.1007/s00382-016-3071-1), which provides an in-depth evaluation of CMIP5 model output against CloudSat and ERA-Interim data. With CMIP6 (and ERA5) output now available and the IPCC AR6 in preparation, this update is timely. This is why we chose the Brief Communication format: We build heavily on previously published and well-recognized work (56 citations in peer-reviewed literature, according to the Web of Science on April 30, 2020), which provides the in-depth background discussion the reviewer seems to be calling for. We clarified this at the end of the introduction in the following sentence:

  > Using new reanalyses and output of the most recent CMIP exercise, this work provides a brief update of the analysis by Palerme et al. (2017), which focused on CMIP5 and ERA-Interim.

**Uncertainties of the CloudSat dataset over some regions**

- Reviewer comment: *While the authors present CloudSat as a benchmark data set, its performance over the Antarctic interior is highly doubtful.*

- Reply: To assess the statistical significance of the difference between the observations from CloudSat and the reanalyses and the models,

we performed a Welch t-test that has been added in Appendix D. As it was mentionned, it attests that the difference between the snowfall means of the CloudSat dataset and ERA5 and the CMIP (5 & 6) ones are significantly different in some regions. In particular, comparisons have to be taken with caution on the Plateau and the Low West regions where there may be a significant underestimation of the surface snowfall means by CloudSat - due to various microphysical processes occuring at lower levels. We choose to keep the results of the CloudSat climatology as it is the only source of observations covering most of the Antarctic continent but we highlighted that there is a low confidence in these areas.

**Link between temperature and precipitation**

- Reviewer comment: *As the authors surely can confirm, there are many levels of complexity involved with precipitation formation, and only focusing on near-surface temperature definitely understates these complexities.*

- Reply: Of course, as the reviewer rightly presumes, we did not intend to imply that near-surface temperature was the only parameter or process that determines Antarctic precipitation rates, even though there is a strong link between temperature and precipitation changes over the Antarctic Ice Sheet on long time scales and larger spatial scales, as ample literature on this issue shows. But of course we agree that this is at best a very first order effect, with many much more subtle physical processes involved in precipitation formation. These processes complicate the detailed picture as soon as one goes beyond the very first order. To reduce the risk of such misunderstandings among our readers, we decided to 1) remove the description of the temperature station data from the main text and to move it into the supplementary material) and, more importantly, to 2) rewrite the short paragraph mentioning the reduced temperature errors:

  > We note that although there is no progress in the representation of large-scale mean precipitation and of its seasonality from CMIP5 to CMIP6, there is a concomitant measurable progress in the representation of surface air temperature. Regional-scale multi-model median root-mean square errors are reduced by typically 5 to 10% between these successive CMIP generations (see Figure D1 in the annex). This indicates that in spite of a clear physical link between temperature and precipitation changes on long time scales (e.g., Krinner et al., 2008;

Frieler et al., 2015), precipitation errors in current-generation AGCMs are not dominated by the first-order physical link between temperature and water vapour saturation pressure, but by errors in the representation of other processes such as atmospheric circulation and cloud microphysics.

**Temperature analysis**

- Reviewer comment: *The statistical analysis of near-surface temperature needs to be expanded. Only the RMSE is currently shown, but that fails to represent the mean bias ... Moreover, there is no analysis of statistical significance whatsoever ...*

- Reply: The mean bias has no sense in this analysis because of the risk or error compensation. We compute the error statistics using monthly means over the entire mean annual cycle. Therefore, one could, for example, obtain a zero annual mean bias in spite of a strong positive summer bias, compensated for by an equally strong, but opposite, winter bias. The RMSE (or, alternatively, the mean absolute bias) is therefore much more meaningful. To allow for a visual assessment of the reality of the improvement, we added error bars indicating the regional mean inter-model standard deviation of the simulated temperature errors, which shows reduced spread in CMIP6 compared to CMIP5 and, depending on the regions, substantial improvement of the mean root mean square errors. As this short analysis of temperature errors is not the focus of the paper and only very riefly mentioned in the main text, we do not think that a more detailed analysis of this aspect is warranted.

**Map of the regions**

- Reviewer comment: *One smaller issue is that Figure 1 should mask out regions south of 82 South. What are the white areas.*

- Reply: The map of the studied regions (Figure 1) has been modified to mask out regions south of 82° South where there is no data from CloudSat.